# Engineered Mesenchymal Stem Cells Over-Expressing BDNF Protect the Brain from Traumatic Brain Injury-Induced Neuronal Death, Neurological Deficits, and Cognitive Impairments

**DOI:** 10.3390/ph16030436

**Published:** 2023-03-13

**Authors:** Bo Young Choi, Dae Ki Hong, Beom Seok Kang, Si Hyun Lee, Seunghyuk Choi, Hyo-Jin Kim, Soon Min Lee, Sang Won Suh

**Affiliations:** 1Department of Physical Education, Hallym University, Chuncheon 24252, Republic of Korea; 2Institute of Sports Science, Hallym University, Chuncheon 24252, Republic of Korea; 3Department of Pathology and Laboratory Medicine, Emory University School of Medicine, Atlanta, GA 30322, USA; 4Department of Physiology, Hallym University College of Medicine, Chuncheon 24252, Republic of Korea; 5SL BiGen, Inc., SL BIGEN Research Hall, 85, Songdogwahak-ro, Yeonsu-gu, Incheon 21983, Republic of Korea

**Keywords:** brain-derived neurotropic factor, stem cells, mesenchymal stem cell administration, traumatic brain injury, neuron death, neurogenesis

## Abstract

Traumatic brain injury (TBI) causes transitory or permanent neurological and cognitive impairments, which can intensify over time due to secondary neuronal death. However, no therapy currently exists that can effectively treat brain injury following TBI. Here, we evaluate the therapeutic potential of irradiated engineered human mesenchymal stem cells over-expressing brain-derived neurotrophic factor (BDNF), which we denote by BDNF-eMSCs, in protecting the brain against neuronal death, neurological deficits, and cognitive impairment in TBI rats. BDNF-eMSCs were administered directly into the left lateral ventricle of the brain in rats that received TBI damage. A single administration of BDNF-eMSCs reduced TBI-induced neuronal death and glial activation in the hippocampus, while repeated administration of BDNF-eMSCs reduced not only glial activation and delayed neuronal loss but also enhanced hippocampal neurogenesis in TBI rats. In addition, BDNF-eMSCs reduced the lesion area in the damaged brain of the rats. Behaviorally, BDNF-eMSC treatment improved the neurological and cognitive functions of the TBI rats. The results presented in this study demonstrate that BDNF-eMSCs can attenuate TBI-induced brain damage through the suppression of neuronal death and increased neurogenesis, thus enhancing functional recovery after TBI, indicating the significant therapeutic potential of BDNF-eMSCs in the treatment of TBI.

## 1. Introduction

Traumatic brain injury (TBI) is a major brain injury, globally affecting various populations in a non-selective manner [1], with approximately 40–50 million affected by the TBI-induced social/financial burden and leading to 4.7 million deaths annually [2]. TBI can devastate the local brain area, contributing to neuronal damage and subsequent cognitive impairment. Previous studies have been conducted to understand TBI’s pathogenesis and ameliorate its symptoms, concentrating on targeting several aspects of the secondary brain damage cascade. Nevertheless, most studies focused on TBI have yielded limited translational accomplishments [3]; likewise, trials utilizing hypothermia and decompressive craniotomy to reduce brain damage have presented limited results.

Stem cell therapy or stem cell-based gene therapy have risen as promising strategies in diverse neuronal injuries that do not yet have any effective outcomes, including Alzheimer’s disease (AD) [4], Parkinson’s disease (PD) [5], amyotrophic lateral sclerosis (ALS) [6], Huntington’s disease [7], stroke [8], and traumatic brain injury [9]. Diverse categories of stem cells involving mesenchymal stem cells (MSCs), neural stem cells (NSCs), multipotent adult progenitor cells (MAPCs), and endothelial progenitor cells (EPCs) have been successfully used to target brain disorders [10,11]. One of the stem cells, MSCs, has therapeutic effects on several brain diseases, where MSCs recover brain damage via upregulation of para- and autocrine factors; it enhances cellular differentiation and programmed cell death process [12,13]. Nevertheless, no successful stem cell therapy has been reported for treating TBI. Brain-derived neurotrophic factor (BDNF) has essential roles in regulating the neurogenesis process: neural differentiation, maturation, and survival [14,15,16]. BDNF delivery has been shown to be very useful in animal models of neurological disorders [17,18,19,20,21]. MSCs have recently been used as a platform of BDNF transportation for treating neurological disorders, including Huntington’s disease and ischemic stroke [22,23]. However, despite its promising potential as a treatment tool for various brain diseases, efforts to use BDNF in the treatment of neurological diseases have complicated problems of application because BDNF has a short half-life (within 10 min) and is unable to cross the blood-brain barrier (BBB) due to its large size (27 kDa) [24]. In order to overcome these limitations, we have previously engineered naïve human MSCs to over-express BDNF. By using these BDNF over-expressing engineered MSCs after irradiation, we confirmed their efficacy (both in vitro and in vivo) in facilitating recovery from neonatal hypoxic-ischemic encephalopathy (HIE) in a rodent model [25].

In order to overcome the limitation of BDNF application, we developed and evaluated the therapeutic approach of irradiated BDNF-eMSCs in protecting the brain against neuronal death, neurological deficits, and cognitive impairment that arise following TBI. We consider two main objectives: (1) to determine the optimal concentration for BDNF-eMSC administration and (2) to compare the efficacy between single and repeated administrations of BDNF-eMSCs. We observed that repeated administrations of BDNF-eMSCs (5 × 10^5^ cells per administration) into the cerebroventricular cavity of rats after TBI led to significant protective effects against neuronal death, neurological deficits, and cognitive impairment following TBI.

## 2. Results

### 2.1. In Vitro Cellular Characterization and Neural Protection Effect of BDNF-eMSCs

Although cellular senescence of naïve MSCs by extensive in vitro passage induces changes in cell morphology as well as proliferation, immortalized cells are free from cellular senescence, and we previously reported this property using non-irradiated BDNF-eMSCs [25]. Long-term culturing of naïve MSCs and non-irradiated BDNF-eMSCs (about 20 passages) demonstrated that the cell shape of the naïve MSCs gradually enlarged with passage, whereas the cell shape of non-irradiated BDNF-eMSCs was well-maintained until passage 20 (Appendix A). As these characteristics challenge the safety of immortalized cells, we carried out an in vitro transformation assay using BDNF-eMSCs before in vitro and in vivo TBI experiments in order to ensure the safety of the BDNF-eMSCs. Unlike the positive control HeLa cell, BDNF-eMSCs did not show transformation activities (Appendix A).

The BDNF-eMSCs have been genetically engineered to express human BDNF protein, and the BDNF protein expression levels of naïve MSCs and BDNF-eMSCs were analyzed using ELISA. Supernatants from naïve MSCs and BDNF-eMSCs were analyzed, and about 258 ng/mL BDNF protein was measured in BDNF-eMSCs, while only 1 ng/mL BDNF protein was measured in naïve MSCs under certain culture conditions (Naïve MSCs, 1.24 ± 0.02; BDNF-eMSCs, 258.09 ± 48.33; Average number, Figure 1A).

In the cellular signaling pathway, BDNF binds to the Tropomyosin receptor kinase B (TrkB) receptor present on the cell membrane. The TrkB receptor is widely expressed in the brain; for example, in the cortex, hippocampus, and multiple brain stem and spinal cord nuclei [26]. The BDNF-bound TrkB receptor undergoes dimerization, the phosphorylation of intracellular residues, and the activation of various intracellular signaling pathways. To confirm that BDNF-eMSCs activate the intracellular signaling of TrkB-expressing cells, we treated retinoic acid-induced differentiated SH-SY5Y cells with the supernatant of BDNF-eMSCs, and confirmed the activation of intracellular signaling pathways, including TrkB, Phospholipase C Gamma (PLC-γ), AKT serine/threonine kinase (AKT), and extracellular-signal-regulated kinase ERK; however, treatment with recombinant TrkB (TrkB-hyFc) diminished BDNF-dependent activation signaling (Figure 1B).

In order to test whether the secreted BDNF protein containing the supernatant of BDNF-eMSCs had any effect in the in vitro TBI mimic model, we performed an in vitro scratch assay using mouse primary hippocampal neurons. In the group of primary neurons co-cultured with BDNF-eMSCs (1:1 ratio to neuron) using trans-well, cell viability was increased, and compared to the scratch-only group; in contrast, this increase was not observed in the recombinant TrkB-hyFc protein (700 ng/mL) treatment group (Control, 100.00 ± 0.00; scratch-only, 10.33 ± 3.03; BDNF-eMSCs, 58.30 ± 6.42; BDNF + rhTrkB-hyFc 25.00 ± 10.27; Average number, Figure 1C). These results indicate that the secreted BDNF protein of BDNF-eMSCs activates BDNF-TrkB intracellular signaling and rescues cell viability under physical stress conditions.

### 2.2. Proteomic Analysis of Supernatant of BDNF-eMSCs

To analyze the paracrine factors of BDNF-eMSCs, including BDNF, that may contribute to treating traumatic brain injury, we performed proteomic analysis of BDNF-eMSCs and naïve MSCs using conditioned media. Proteomic analysis by LC-MS/MS revealed significant changes to the total proteome and secretome in BDNF-eMSCs over naïve MSCs. Principal component analysis (PCA) and correlation analysis indicated a separation of naïve MSCs and BDNF-eMSCs in the total proteome and secretome (Figure 2A,B), demonstrating that BDNF-eMSCs have a distinct protein expression profile compared to naïve MSCs. By filtration, we identified significantly up- or down-regulated differentially expressed proteins (DEPs) in BDNF-eMSCs, compared to naïve MSCs (Figure 2C,D). Next, we performed gene ontology analysis to define biological processes of the up-regulated BDNF-eMSC secretome, and the results showed that the expression of numerous proteins known to be involved in neural regeneration or metabolism was increased in BDNF-eMSCs (Figure 2E). These data suggest that the BDNF-eMSC supernatant may contain many factors, in addition to BDNF, that may be involved in the TBI treatment mechanism.

### 2.3. BDNF-eMSC Administration Reduced Neuronal Death and Glial Activation after TBI

Based on the above results, we sought to assess the effects of BDNF-eMSCs on neuronal loss and glial activation in an animal model of TBI. Rats were subjected to CCI-induced TBI (Figure 3A), and then different low–high dosages of BDNF-eMSCs (i.e., 1 × 10^5^, 5 × 10^5^, or 1 × 10^6^ cells) were administered via ICV 24 h after TBI, in order to determine the optimal concentration for BDNF-eMSC administration (Figure 3B,C). Brains were harvested 2 weeks after TBI, and we conducted immunocytochemistry analyses (Figure 3D). First, we tested whether the administration of BDNF-eMSCs reduced TBI-induced neuronal death in the ipsilateral hippocampus using NeuN, a neural cell marker. After TBI, the vehicle-treated group showed weak staining for NeuN^+^ neurons in the hippocampal CA1. There were no significant differences in the number of NeuN^+^ neurons between vehicle- and 1 × 10^5^ BDNF-eMSC-treated groups; however, the number of NeuN^+^ neurons in both 5 × 10^5^ and 1 × 10^6^ BDNF-eMSC-treated groups was significantly higher than in the vehicle or 1 × 10^5^ BDNF-eMSC-treated groups, although there were no statistically significant differences in the number of NeuN^+^ neurons between the 5 × 10^5^ and 1 × 10^6^ BDNF-eMSC treatment groups (TBI-Vehicle, 30.90 ± 7.14; TBI-1 × 10^5^ BDNF-eMSC, 39.10 ± 7.31; TBI-5 × 10^5^ BDNF-eMSC, 88.10 ± 9.04; TBI-1 × 10^6^ BDNF-eMSC, 107.75 ± 18.39; average number; Figure 4A,B).

Previous studies have suggested that microglia and astrocyte activation generally occurs after TBI [27]. These activated glial cells can give rise to detrimental neurotoxic effects by releasing numerous cytotoxic substances, such as oxidative metabolites and proinflammatory cytokines [28]. We found that TBI induced by CCI increased microglial activation, as evidenced by switching from a highly ramified to an amoeboid shape, as indicated by a significant increase in the fluorescence intensity of Iba-1 (an activated microglia marker). However, administration of BDNF-eMSCs at 5 × 10^5^ or 1 × 10^6^ cells significantly attenuated the fluorescence intensity of Iba-1 (TBI-Vehicle, 11,111.84 ± 1933.67; TBI-1 × 10^5^ BDNF-eMSC, 8252.12 ± 1569.85; TBI-5 × 10^5^ BDNF-eMSC, 4520.20 ± 172.75; TBI-1 × 10^6^ BDNF-eMSC, 4494.36 ± 285.09; average gray-scale intensities; Figure 4A,C) in the ipsilateral CA1 after TBI, compared with vehicle- and 1 × 10^5^ BDNF-eMSC-treated groups.

In addition, we also found that TBI led to dramatic pathological changes in astrocytes, such as reactive gliosis and glial scar formation, as indicated by a significant increase in the fluorescence intensities of S100B, GFAP, and AQP4. However, the administration of BDNF-eMSCs at 5 × 10^5^ or 1 × 10^6^ cells significantly reduced the fluorescence intensity of S100B (TBI-Vehicle, 12,156.84 ± 280.70; TBI-1 × 10^5^ BDNF-eMSC, 11,930.95 ± 337.73; TBI-5 × 105 BDNF-eMSC, 7999.91 ± 567.71; TBI-1 × 10^6^ BDNF-eMSC, 7527.52 ± 660.24; average gray-scale intensities; Figure 4D,E), GFAP (TBI-Vehicle, 19,162.90 ± 1297.10; TBI-1 × 10^5^ BDNF-eMSC, 17,522.50 ± 840.15; TBI-5 × 10^5^ BDNF-eMSC, 7375.57 ± 746.19; TBI-1 × 10^6^ BDNF-eMSC, 7527.58 ± 51.31; average gray-scale intensities; Figure 4D,F) and AQP4 (TBI-Vehicle, 20,336.68 ± 1806.37; TBI-1 × 10^5^ BDNF-eMSC, 18,300.99 ± 1344.47; TBI-5 × 10^5^ BDNF-eMSC, 4930.59 ± 1379.91; TBI-1 × 10^6^ BDNF-eMSC, 5981.73 ± 149.52; average gray-scale intensities; Figure 4D,G) in the ipsilateral CA1 after TBI, compared with vehicle- and 1 × 10^5^ BDNF-eMSC-treated groups. These results suggest that BDNF-eMSC administration reduces neuronal death and attenuates glial activation after TBI.

### 2.4. Long-Term Neuroprotective Effects of Repeated BDNF-eMSC Administration against TBI

To further clarify the translational approach to this research, we studied the therapeutic efficacy of repeated administration of BDNF-eMSCs post-TBI. BDNF-eMSCs were administered every two weeks, starting 24 h after TBI and up to 8 weeks post-injury, to compare single and repeated administrations of BDNF-eMSCs (Figure 3E). First, we measured the TBI-induced lesion area in cresyl violet-stained coronal brain sections 8 weeks after TBI (Figure 5A). Single or repeated administrations of BDNF-eMSCs significantly reduced the TBI-induced lesion area compared with the vehicle-treated group. In contrast, there was no significant decrease in lesion cavity area in the MSC-treated group compared to the vehicle-treated group (Figure 5B).

Next, we tested whether repeated administration of BDNF-eMSCs also provided long-term neuroprotective effects and attenuated astroglial activation in the ipsilateral hippocampus at 8 weeks post-TBI (Figure 5C). Sham-operated groups showed intense staining for NeuN^+^ neurons in the hippocampal CA1. Compared with the sham group, the number of NeuN^+^ neurons decreased at 8 weeks after TBI. However, the number of NeuN^+^ neurons in the single and repeated BDNF-eMSC-treated groups was significantly higher than in the vehicle-treated group. In addition, repeated BDNF-eMSC administration led to a significant increase in the number of NeuN^+^ neurons following TBI, compared with the single BDNF-eMSC-treated group. In contrast, the MSC-treated group clearly indicated no significant difference in the number of NeuN^+^ neurons when compared with the vehicle-treated group (Sham, 172.97 ± 6.26; TBI-Vehicle, 8.62 ± 0.97; TBI-BDNF-eMSC-S, 64.88 ± 8.03; TBI-BDNF-eMSC-R, 102.38 ± 14.88; TBI-MSC, 15.20 ± 4.49; average number; Figure 5D). We also evaluated the effects of repeated BDNF-eMSC administrations on astroglial activation after TBI. After TBI, GFAP fluorescence intensity significantly increased in the CA1 region of the hippocampus; however, in the case of the repeated BDNF-eMSC-treated group, a significant reduction of GFAP fluorescence intensity was observed, compared with the vehicle-, single BDNF-eMSC-, and MSC-treated groups (Sham, 3710.92 ± 128.54; TBI-Vehicle, 8984.16 ± 378.88; TBI-BDNF-eMSC-S, 8542.02 ± 227.43; TBI-BDNF-eMSC-R, 5073.48 ± 168.97; TBI-MSC, 8514.24 ± 304.15; average gray-scale intensities; Figure 5E). These results demonstrate that repeated administration of BDNF-eMSCs provides a long-term neuroprotective effect and decreases astroglial activation after TBI.

### 2.5. Repeated BDNF-eMSC Administration Increased Adult Hippocampal Neurogenesis and Gliogenesis after TBI

To assess whether repeated administration of BDNF-eMSCs influenced adult hippocampal neurogenesis after TBI, we birth-dated newborn cells by BrdU injection twice daily for 4 consecutive days from the third day following TBI. We investigated their fate in the sub-granular zone (SGZ) and granular cell layer (GCL) at 8 weeks after TBI (Figure 6A). Surviving newborn cells, as detected by BrdU immunostaining, were distributed throughout the entire GCL at 8 weeks after sham surgery. Some cells had already migrated into the GCL, whereas others were still localized in the SGZ. The quantification of BrdU^+^ cells indicated a significant increase in the number of BrdU^+^ cells in the SGZ/GCL after TBI. The number of BrdU^+^ cells from rats repeatedly administrated BDNF-eMSCs was significantly greater than those in sections from vehicle-, single BDNF-eMSC-, or MSC-treated groups (Sham, 7.63 ± 1.24; TBI-Vehicle, 67.87 ± 4.76; TBI-BDNF-eMSC-S, 71.04 ± 6.13; TBI-BDNF-eMSC-R, 99.60 ± 5.99; TBI-MSC, 61.40 ± 5.87; average number; Figure 6B). We then explored the phenotypes of cells that had survived in the SGZ/GCL, as detected by triple immunofluorescence staining with BrdU, NeuN, and GFAP. When calculating the number of adult newborn neurons or glia, we found that repeated administration of BDNF-eMSCs only revealed a remarkable increase in the total number of BrdU^+^ cells double-labeled for the specific marker of mature neurons (NeuN) or mature astrocytes (GFAP) in the SGZ/GCL, as compared to vehicle-treated groups (BrdU^+^NeuN^+^ cells; Sham, 4.83 ± 1.01; TBI-Vehicle, 13.71 ± 1.48; TBI-BDNF-eMSC-S, 28.80 ± 6.50; TBI-BDNF-eMSC-R, 40.40 ± 4.07; TBI-MSC, 19.48 ± 3.16; BrdU^+^GFAP^+^ cells; Sham, 2.23 ± 0.28; TBI-Vehicle, 19.31 ± 1.55; TBI-BDNF-eMSC-S, 39.16 ± 7.75; TBI-BDNF-eMSC-R, 54.63 ± 5.57; TBI-MSC, 39.48 ± 5.26; average number; Figure 6C,D). In addition, BDNF-eMSCs revealed an increase in the percentage of BrdU^+^NeuN^+^ cells among BrdU^+^ cells in both single and repeated administrations, compared with vehicle- or MSC-treated groups. BDNF-eMSCs also increased the percentage of BrdU^+^GFAP^+^ cells among BrdU^+^ cells in both single and repeated administrations, compared with the vehicle-treated group, although the MSC-treated group showed a much higher percentage of BrdU^+^GFAP^+^ cells among BrdU^+^ cells than single or repeated BDNF-eMSC-treated groups (Figure 6E). Thus, the obtained results demonstrate that BDNF-eMSCs reinforce neurogenesis and gliogenesis in the adult hippocampus after TBI.

### 2.6. Repeated BDNF-eMSC Administration Improved Neurological and Cognitive Function after TBI

Finally, we examined the effects of repeated administrations of BDNF-eMSCs on neurological performance and cognitive abilities through the mNSS test and MWM. These tests specifically evaluate motor- and memory-related brain regions, such as the hippocampus and cortex, which are often compromised after TBI. Our results indicated that repeated administration of BDNF-eMSCs led to remarkable protective effects on neurological function, as demonstrated by a decrease in the mNSS score (Figure 7A) and an increase in ΔmNSS scores (Figure 7B) compared with other groups. We also found that multiple treatments with BDNF-eMSCs resulted in improved cognitive ability, as indicated by a significant reduction in escape latency (Figure 7C) and increased platform crossings (Figure 7D) compared to other groups. However, compared to other groups, there were no significant differences in the time spent in the target quadrant and velocity (Figure 7E,F). In addition, Figure 7 shows improvement in mNSS at 35 days and in the MSC-treated group’s MWM escape latency. Although naïve MSCs appear to have limited therapeutic potential and BDNF is only partially responsible for the animals’ cognitive recovery, these results demonstrate that the repeated administration of BDNF-eMSCs reverses TBI-induced neurological and cognitive disabilities.

## 3. Discussion

In this study, we analyzed the therapeutic potential of BDNF-eMSCs using in vitro and in vivo models of TBI. Our results demonstrated that BDNF-eMSCs secrete significantly more BDNF compared to naïve MSCs. Consequently, BDNF-eMSC administration reduced neuronal death, microglial activation, and astroglial activation in the hippocampus after TBI. Furthermore, repeated administration of BDNF-eMSCs reduced glial activation and delayed neuronal loss, enhanced neurogenesis in rats with TBI, improved neurological and cognitive functional recovery, and reduced the lesion area in the damaged brain of rats with TBI. Together, these results demonstrate that BDNF-eMSCs attenuate TBI-induced brain damage by reducing neuronal death and increasing neurogenesis.

Brain edema and swelling induced by increased intracranial pressure (ICP) are the most significant causes of mortality in TBI patients [29]. Medical treatment of intracranial hypertension is typically regulated by combining sedative and osmotic agents, such as hypertonic saline and mannitol [30]; however, a pharmacological agent targeting the specific pathologies of TBI-induced neuronal death remains unidentified. In TBI, the mRNA of BDNF expression is increased transiently and significantly in the injured hippocampus and cortex [31], with the BDNF level declining at 24 h post-injury and is no longer changed at 36 h post-injury [32]. TrkB receptor mRNA is also transiently increased in the hippocampus and dentate gyrus following injury [33]. This transient surge of BDNF and TrkB receptor following TBI reveals that BDNF plays an endogenous neuroprotective role, attempting to ameliorate secondary cell damage [34]. BDNF/TrkB signaling cascades in brain function regulation contribute to the therapeutic potential of BDNF/TrkB with respect to several neurological diseases, including TBI. However, the limitations associated with neuroprotective BDNF include its short half-life (<10 min) and inability to cross the blood-brain barrier (BBB) due to its large size (27 kDa) [24]. To date, the direct application of BDNF for ameliorating TBI-induced brain damage has not been identified. Several studies have demonstrated the beneficial effects of genetically modified MSC-based therapies for brain injury. MSCs are emerging as an attractive approach for restorative medicine in central nervous system (CNS) diseases and injuries, such as TBI, due to their relatively easy derivation and therapeutic effect following administration. In addition, MSCs are considered a safer approach for gene therapy. Over-expressing neuroprotective genes such as BDNF provide the dual benefit of promoting the homing and survival of transplanted MSCs, as well as the paracrine factor-induced recovery and neuroprotection of the host’s brain-injured area. A recent study has shown that using genetically modified MSCs to over-express BDNF mitigated hypoxic ischemic (HI) deficit by reducing brain infarct, preventing apoptosis, and reducing astrogliosis and inflammatory responses in a rat model of severe neonatal HI brain injury [25]. Furthermore, Yuan et al. have demonstrated that MSCs isolated from the umbilical cord and genetically modified to over-express BDNF increased their ability to migrate to and survive in cerebral tissues and mitigated neurological deficits more efficiently than MSCs alone in rats with TBI [35]. In this paper, we also observed that the administration of BDNF-eMSCs reduced TBI-induced pathological outcomes such as neuronal death, neurological deficits, and cognitive impairment.

Excessive glial activation due to TBI also contributes to further neuronal injuries. Microglia are the major glial cells in the brain, which help to maintain the homeostasis of the central nervous system through the secretion of neurotrophic factors. However, the excessive and rapid activation of microglia leads to secondary neuronal damage through pro-inflammatory and immune responses after neurological insults, such as TBI [36]. Activated microglia initiate morphological changes into an amoeboid pattern, migrate to the injured site, and release several neurotoxic substances such as RNS/ROS, pro-inflammatory cytokines, and metalloproteinases (MMPs) [37,38]. S100B, an acidic calcium-binding protein, is known to be an astrocyte-specific marker protein similar to GFAP. Up-regulation of S100B protein synthesis and leakage of S100B from damaged astrocytes that express GFAP in the glial scar can be induced by acute brain injury (e.g., stroke or TBI). These events give rise to an increase in the pathogenic process. This study found that the administration of BDNF-eMSCs significantly decreased glial activation.

The hippocampus is one of two prominent brain regions in which neurogenesis takes place, and this region is well-defined as the warehouse of learning and memory. BDNF is a key molecule that regulates neuronal differentiation and survival, synaptic plasticity, as well as activity-dependent changes in synaptic structure and function [39]. Long-term potentiation (LTP) is a specific form of synaptic plasticity that occurs in the brain, which is the cellular basis of learning and memory. BDNF is a major regulator for forming and maintaining LTP in the hippocampus and other brain regions [40,41]. Studies have established that adult hippocampal neurogenesis is associated with learning and memory functions [42,43]. BDNF and TrkB signaling influence adult neurogenesis by mediating neuronal differentiation and the survival of newly generated neurons [44,45,46]. BDNF and TrkB signaling affects hippocampal neurogenesis, contributing to learning and memory functions. During adult hippocampal neurogenesis, neural progenitor cells (NPCs) are generated and propagated from the SGZ, a region that is located along the border between the granular cell layer (GCL) of the dentate gyrus (DG) and the hilus. To become functional neurons, NPCs must pass through several steps of maturation involving cell proliferation, migration, differentiation, and survival. However, not all newly produced cells become neurons: A sub-population develops into glial cells, and a substantial number die before complete maturation. Several studies have found that neurons account for most surviving newborn cells, with proportions ranging between 60% [47] and 93% [48]. Growing evidence has suggested that brain injury transiently increases the number of NPCs in rodent and primate SGZ. Despite the increase in the number of NPCs after brain injury, these cells prematurely die without the ability to repair the injured CNS. In the present study, we found that the neuronal phenotype of the newly generated cells after TBI was significantly decreased, when compared to the sham-operated group. However, the repeated administration of BDNF-eMSCs induced a remarkable increase in the number of newly generated cells compared to the vehicle-, single BDNF-eMSC-, or MSC-treated groups. We also found that the number of newly generated cells with neuronal phenotype in the repeated BDNF-eMSC group after TBI was significantly higher compared to vehicle- and MSC-treated groups.

Monotarget therapy for TBI is generally not effective due to the multi-factorial and heterogeneous nature of TBI, given that various manifestations occur in different parts and time points post-injury. Therefore, an ideal therapeutic strategy should utilize a multi-target, simultaneous action to induce a robust treatment for TBI [49]. One promising therapeutic option with multi-target, simultaneous action is stem cell-based gene therapy due to their secretion of neurotrophic factors and other neuroprotective factors [50]. In the present study, we performed proteomic analysis to identify the paracrine factors of BDNF-eMSCs and naïve MSCs, which were used as a control to confirm the change in secretome profile due to genetic engineering. Compared to the naïve MSCs, significantly up- and down-regulated proteins in BDNF-eMSCs were identified (Appendix A). Among the up-regulated proteins (including BDNF, as expected), a number of them have been reported to show neuroprotective functions in the brain. In particular, it has been reported that metalloproteinase inhibitor 1 (TIMP) and agrin (ARGN) have neuroprotective and recovery functions in animal TBI models [51,52]. Through proteomic analysis of the secretome expressed by BDNF-eMSCs, it was confirmed that other effective molecules could act alongside BDNF in the treatment of TBI.

This study potentially has several limitations that must be addressed in future studies. (1) In the present study, we did not verify the presence of BDNF-eMSCs by histological analysis. After irradiation, we administered BDNF-eMSCs to the animals and expected them to present in one or two weeks. After BDNF-eMSCs were administered to normal mice, the residual amount was measured by qPCR. We found that BDNF-eMSCs disappeared from the brain within a week. It would be interesting to explore the presence of BDNF-eMSCs for a longer period after administration. (2) Exploring the mechanism of BDNF-eMSCs after TBI may be a future direction for human translation.

## 4. Materials and Methods

### 4.1. Cell Preparation

The establishment of the engineered human mesenchymal stem cells over-expressing BDNF has been described in the previous report [25]. To summarize, they present a homogenous spindle-shaped cell morphology, express common MSC surface marker proteins, have robust cell proliferation properties, and stably secrete BDNF during long-term culture. Before conducting the in vitro and in vivo efficacy tests, the cells were irradiated with 200 Gy radiation using an X-ray irradiation device (Rad Source Technologies, Buford, GA, USA) in order to prevent proliferation. In the following, these irradiated cells are denoted as BDNF-eMSCs. BDNF-eMSCs were stored in the LN_2_ vapor phase under freezing conditions until used for in vitro and in vivo experiments. BDNF-eMSCs were thawed, prepared through a concentration procedure, and kept at 2~8 °C until used for experiments.

### 4.2. Analysis of Cell Morphology

To determine the morphological maintenance of BDNF-eMSCs compared to the naïve MSCs, 0.6 × 10^6^ cells of BDNF-eMSCs and 1 × 10^6^ cells of naïve MSCs, on average, were seeded into T175 flasks (Thermo Fisher Scientific, Waltham, MA, USA). After 3–4 days, cell morphology was observed by optical microscope (NIKON, Tokyo, Japan), and images were captured (original magnification 100×).

### 4.3. In Vitro Transformation Assay

An in vitro cell transformation assay was performed to confirm the non-transforming property of the BDNF-eMSCs. The colony-formation ability was measured using a CytoSelectTM 96-well Cell Transformation Assay kit (Cell Biolabs, Inc., San Diego, CA, USA), according to the manufacturer’s instructions. Purchased HeLa and NIH3T3 cells (Korea Cell Line Bank, Seoul, Korea) were used as positive and negative controls, respectively. BDNF-eMSCs and control cells were seeded at a density of 5 × 10^4^ cells per 96-well plate (Corning) with an agar matrix. After 8 days at 37 °C in a 5% CO_2_ incubator, agar matrices were dissolved and stained with CellTiter 96^®^ AQueous One Solution Reagent (Promega, Madison, WI, USA) for 4 h. The absorbance of the samples at 490 nm was measured by SpectraMAX 190 microplate reader.

### 4.4. Enzyme-Linked Immunosorbent Assay (ELISA)

Enzyme-linked immunosorbent assay (ELISA) was performed to quantitate BDNF in the culture supernatant. Naïve MSCs and BDNF-eMSCs were seeded at a density of 1 × 10^5^ cells per 12-well plate (Corning, Corning, NY, USA) with 1 mL complete media. Removed cell debris supernatant was obtained after 48 h incubation. BDNF concentrations were analyzed using a Human BDNF ELISA kit (R&D Systems, Minneapolis, MN, USA). The absorbance of the samples at 450 nm was measured using a SpectraMAX 190 (Molecular Devices, San Jose, CA, USA) microplate reader.

### 4.5. Immunoblot Analysis

Immunoblot was performed to confirm the intracellular signaling pathway through the BDNF–TrkB pathway. To induce differentiation of SH-SY5Y cells (Korea Cell Line Bank), 10 μM of retinoic acid (Sigma-Aldrich, Saint Louis, MO, USA) was used. Recombinant BDNF protein (Peprotech, Rocky Hill, NJ, USA) was treated as positive control and supernatant of BDNF-eMSCs, together or alone with recombinant human TrkB-hyFc (Sino Biological, Beijing, China), were incubated with differentiated SH-SY5Y cells for 1 h at the 37 °C, 5% CO_2_ incubator. BDNF-eMSCs were harvested and lysed with RIPA buffer (Thermo Fisher Scientific). Lysate proteins were distinguished using SDS-PAGE, transferred to nitrocellulose membranes (Thermo Fisher Scientific), and incubated with blocking buffer (Thermo Fisher Scientific) for 30 minutes. Immunoblots were performed using primary antibodies against TrkB, PLC-γ, AKT, ERK, Phospho-TrkB (Tyr706/Tyr707), Phospho-PLC-γ (Tyr783), Phospho-AKT (Ser473), Phospho-ERK (Thr202/Tyr204) (Cell Signaling, Danvers, MA, USA), and ACTIN (MP Biomedicals, Santa Ana, CA, USA). Horseradish peroxidase (HRP)-conjugated mouse (GeneTex, Irvine, CA, USA) and rabbit (Cell signaling) secondary antibodies were detected with ECL reagent (Thermo Fisher Scientific) and a ChemiDoc XRS+ System (Bio-Rad, Hercules, CA, USA).

### 4.6. In Vitro Scratch Assay

E18 primary hippocampal neurons were seeded into a Poly-L-lysine coated 24-well plate at 5 × 10^5^ cells/mL and grown for 14 days at 37 °C under 5% CO_2_ in Neurobasal/B27 culture medium. After 14 days, a wound was formed by scratching the cell monolayer in a straight line with a sterile p-200 pipette tip. Survival of the cultured mouse hippocampal neurons after scratching was evaluated by counting the number of living versus dead neurons using a WST based on an EZ-Cytox kit (Dogenbio, EZ-1000, Seoul, Republic of Korea).

We measured the absorbance of each sample against an empty area as background control using a microreader. The wavelength for analyzing the absorbance of the formazan product ranged from 420–480 nm.

### 4.7. Proteomic Analysis and Data Processing

To obtain the cell supernatant for proteomic analysis, naïve MSCs and BDNF-eMSCs were seeded with 1 × 10^6^ cells into T175 flasks. After 3 days, the complete medium was changed to phenol-free DMEM (WELGENE, Gyeongsan, Republic of Korea) without serum and cultured for 24 h. All proteomic analysis and data processing were performed at BERTIS INC. The MS/MS raw data were converted to mzML files using MSConvert [53]. The mzML files were searched against a protein database consisting of the SwissProt human reference database (released May 2022) and common contaminants using Comet [54]. The search parameters were semi-tryptic cleavage, number of missed cleavages ≤ 2, and a precursor and fragment mass tolerance of 10 ppm. Carbamidomethylation on cysteine and TMT10plex on lysine and peptide N-terminus were used for static modifications, while oxidation on methionine was used as a variable modification. The search results were filtered by target-decoy analysis with a false discovery rate (FDR) of 1% at peptide spectrum match (PSM) and peptide level. The identified proteins formed a total proteome. Only the proteins reported as secreted proteins in the reference database were included in the secretome.

### 4.8. Differential Expression Analysis

We defined differentially expressed proteins (DEPs) by applying an integrative statistical method previously reported [55]. In brief, we calculated test statistics for each protein using Student’s *t*-test, Wilcoxon Rank-sum test, and a log_2_ median ratio in each comparison. We then estimated empirical distributions of the test statistics and log_2_ median ratios for the null hypothesis by randomly permutating all samples 1000 times. Using the estimated empirical distributions for each protein, we computed adjusted *p*-values for the observed test statistics and log_2_ median ratio, then calculated the overall *p*-value by combining these *p*-values using Stouffer’s method [56]. Finally, we identified DEPs as those with *t*-test *p*-values < 0.05 and absolute log_2_ median ratios greater than log_2_(1.5)-fold in each comparison.

### 4.9. Selection of Signature Proteins

To obtain the signature proteins defining each pipeline, for each protein, we compared its log_2_-fold change in the pipeline with those in the other pipelines using a previously reported integrative statistical hypothesis testing method [55]. For each comparison, the putative signature proteins were first selected as those with *t*-test *p*-values < 0.05 and absolute log_2_ median ratios greater than log_2_(1.5)-fold through the above hypothesis testing method. We then further filtered the selected proteins by choosing those with (1) the median value of the samples in the pipeline larger than zero and (2) the median value of the remaining samples less than zero.

### 4.10. Gene Ontology Analysis

Gene Ontology analysis was conducted using the BiNGO [57] plug-in in Cytoscape [58]. The Gene Ontology biological processes (GOBP) represented by the signature secretome proteins defining each pipeline were identified as those with *p*-value < 0.05 and the number of molecules involved in the process ≥2. The enriched GOBPs were categorized into “Metabolism” or “Neuronal regeneration” based on their functional relevance. A GOBP network for each pipeline was constructed for the enriched biological processes. The network was constructed by setting nodes as GOBP terms and adding relations among the GOBP terms. The network was drawn using Cytoscape.

### 4.11. Experimental Animals and Ethics Statement

Sprague Dawley male rats (8–10 weeks old, weight 250–350 g; DBL, Chungcheongbuk-do, Republic of Korea) were used in this study. All rats were housed in a gradually maintained room (22 ± 2 °C, 55 ± 5% humidity, 12-h light/dark cycle) and supplied a standard diet (Purina, Gyeonggi-do, Korea) and fresh water. Animal experiments were started following one week of acclimatization to minimize stress during transportation. Experimental procedures were approved (protocol # Hallym 2021–23) in accordance with National Institutes of Health guidelines. This manuscript also complies with the Animal Research: Reporting In Vivo Experiments (ARRIVE) guidelines [59].

### 4.12. In Vivo Model for TBI

We used a controlled cortical impact (CCI) animal model to induce TBI [60,61]. Experimental rats were deeply anesthetized using 3% isoflurane with ventilated 70:30 mixture gas (70:30 ratio, nitrous oxide:oxygen) in an isoflurane device (VetEquip, Livermore, CA, USA). This device was connected with a stereotaxic apparatus (David Kopf Instruments, Tujunga, CA, USA) for anesthetizing gas flow. Approximately 5 mm circle diameter of the cranial skull was isolated during craniotomy (2.8 mm lateral to the midline and 3 mm lambda to the bregma). Then, using a CCI device (Leica Impact One; Leica Biosystems, Nussloch, Germany), a 3 mm flat-tip impactor push down to a 3 mm depth on the dural surface at 5 m/s velocity. After TBI, rats were transferred to a homeothermic incubator for recover body temperature (Harvard Bioscience, Holliston, MA, USA). The TBI-induced pathogenetic outcome was evaluated at 2 and 8 weeks post-injury. Animals were assigned randomly to traumatic brain injury according to an online randomization tool (randomizer.org). All sham groups have only craniotomy except CCI.

### 4.13. BDNF-eMSC Administration and Experimental Design

To investigate the therapeutic efficacy of BDNF-eMSCs following TBI, BDNF-eMSCs were intracerebroventricularly administered directly into the left lateral ventricle of the brain (1.5 mm lateral to the midline and anteroposterior 1.0 mm, mediolateral 1.5 mm, and dorsoventral 3.0 mm from bregma and cortical surface) for 5 min at a rate of 1 µL/min using a syringe pump (Legato 130; KD Scientific, Holliston, MA, USA) and a 10 µL Hamilton syringe with a 29 gauge needle onto the stereotaxic frame. When the infusion was complete, we left the needle to remain in place for an additional 5 min, then removed the needle slowly for 2 min to minimize the backflow of solution out of the injection site. Control rats were injected with equal volumes of saline as the vehicle. The present study had two main objectives. In phase 1, rats were subjected to CCI-induced TBI, and then BDNF-eMSCs were administered 24 h after TBI to determine the optimal concentration for BDNF-eMSC administration. Rats were sacrificed at day 14 post-TBI and divided into 4 groups for histological evaluation: (1) Vehicle-treated TBI group (TBI-veh; *n* = 4), (2) dose 1 BDNF-eMSC-treated TBI group (approximately 1 × 10^5^ cells; *n* = 4), (3) dose 2 BDNF-eMSC-treated TBI group (approximately 5 × 10^5^ cells; *n* = 4), and (4) dose 3 BDNF-eMSC-treated TBI group (approximately 1 × 10^6^ cells; *n* = 4). In phase 2, rats were subjected to CCI-induced TBI, and then BDNF-eMSCs were administered every two weeks, starting 24 h after TBI and up to 8 weeks post-injury, for comparison of single and repeated administrations of the BDNF-eMSCs. In addition, to assess the effects of BDNF-eMSCs in adult hippocampal neurogenesis after TBI, the thymidine analog BrdU (50 mg/kg; Sigma, St. Louis, MO, USA) for labeling newborn cells was intraperitoneally administrated twice per day at 12 h intervals for 4 consecutive days at the third day after TBI. Rats were sacrificed at day 56 post-TBI and divided into 5 groups for functional assessment and histological evaluation: (1) sham-operated group (Sham; *n* = 7), (2) vehicle-treated TBI group (TBI-Veh; *n* = 9), (3) single BDNF-eMSC-treated TBI group (approximately 5 × 10^5^ cells; TBI-BDNF-eMSC-S; *n* = 10), (4) repeated BDNF-eMSC-treated TBI group (approximately 5 × 10^5^ cells; TBI-BDNF-eMSC-R; *n* = 10), and (5) naïve MSC-treated TBI group (approximately 5 × 10^5^ cells; TBI-MSC; *n* = 10).

### 4.14. Tissue Preparation

To obtain brain samples, rats were deeply anesthetized and made unconscious using urethane (1.5 g/kg, intraperitoneal injection) dissolved in saline (0.9% NaCl) at a volume of 0.01 mL/g body weight. A toe pinch was used to confirm the effectiveness of full anesthetization. Rats were intracardially perfused with saline, followed by 4% paraformaldehyde (PFA) in phosphate-buffered saline (PBS). The obtained whole brains were immersed in 4% PFA for post-fixation for 1 h and then transferred into 30% sucrose for cryoprotection for several days. The brain was coronally sectioned thereafter using a cryostat microtome (CM1850; Leica, Wetzlar, Germany) at 30 μm thickness.

### 4.15. Immunofluorescence Analysis

Brain sections were incubated in hydrogen peroxide with methanol for 15 min at room temperature to suppress endogenous peroxidase activity. Then, these sections were incubated with primary antibodies in 0.3% Triton X-100 at a 4 °C incubation overnight. The primary antibodies used in this study were as follows: mouse monoclonal anti-NeuN antibody (1:500; Millipore, Cambridge, UK), goat anti-Iba-1 (1:500; Abcam, Cambridge, UK), goat anti-GFAP (1:1000; Abcam), rabbit anti-aquaporin 4 (AQP4; 1:1000; Cell Signaling Technology, Danvers, MA, USA), and rat anti-BrdU (1:400; Abcam). For double staining, primary antibodies were simultaneously incubated. For NeuN, Iba-1, S100B, GFAP, AQP4, and BrdU, fluorescent-conjugated secondary antibodies were applied (1:250; Invitrogen, Carlsbad, CA, USA). Between incubations, the sections were washed with PBS. Sections were counterstained with DAPI (4,6-diamidino-2-phenylindole; diluted 1:1000; Invitrogen). Immunofluorescence-stained sections were mounted on gelatin-coated slides, and cover slides were mounted with DPX (Sigma-Aldrich). Immunofluorescence signals were detected using a Zeiss LSM 710 confocal microscope (Carl Zeiss, Oberkochen, Germany) with sequential scanning mode for DAPI and Alexa 488, 594, and 647. Stacks of images (1024 × 1024 pixels) from consecutive slices of 0.5–0.8 μm in thickness were obtained by averaging 15 scans per slice, which were processed using ZEN 2 (blue edition, Carl Zeiss). Images were taken from the hippocampal CA1 or dentate gyrus (DG) from the ipsilateral hemisphere. The mean intensity for Iba-1, S100B, GFAP, or AQP4 was quantified by the equivalent area using ZEN 2 blue edition software (Carl Zeiss, Oberkochen, Germany). In addition, five coronal sections (330 μm intervals) related to the bregma were collected from each animal, ranging from 2.92 to 4.56 mm. A blinded observer analyzed the fluorescence images and counted the number of NeuN^+^, BrdU^+^, BrdU^+^GFAP^+^, or BrdU^+^NeuN^+^ cells in the fixed area of hippocampal CA1 or DG from the ipsilateral hemisphere. Regions of interest were drawn manually over the stratum pyramidal (SP) of the CA1 or the granular cell layer (GCL) and SGZ along the superior and inferior blades of the DG. For each animal, immunopositive cells were counted in each section, and the result was expressed as the mean number of immunopositive cells per mm^2^ ± SEM.

### 4.16. Measurement of the Lesion Area

For the determination of the lesion area after TBI, brain sections were stained with cresyl violet and captured using a light microscope (Olympus upright microscope IX70, Olympus, Tokyo, Japan). The lesion area was quantified using coronal sections at 8rostral–caudal levels, spaced at every 330 μm from −2.64 mm to −5.40 mm relative to the bregma. This analysis was performed by a blinded investigator using the ToupView software (ToupTek Photonics, Hangzhou, China).

### 4.17. Assessment of Neurological Deficits

To examine whether the administration of BDNF-eMSCs reduced TBI-induced neurological deficits, we evaluated the modified neurological severity score (mNSS), as described previously [62]. These tests were conducted on day 1 and 1, 2, 3, 4, 5, 6, 7, and 8 weeks following TBI. The mNSS test includes motor (muscle status, abnormal movement), sensory (visual, tactile, and proprioceptive), balance, and reflex trials and is graded from 0 to 18 (0 = normal function, 18 = maximal deficit) [63]. Briefly, this behavioral test is based on: (1) raising the rat by the tail and recording flexion (3 points), (2) walking on the floor (3 points), (3) a sensory test (2 points), (4) beam balance tests (6 points), and (5) reflex absence/abnormal movements (4 points). A point indicates behavioral function inability in the associated task or inadequacy of the tested reflex; thus, a higher score implies a more severe injury.

### 4.18. Morris Water Maze Test

To verify whether BDNF-eMSC administration rescues TBI-induced cognitive decline, rats underwent the Morris water maze (MWM) test starting 50 days after TBI. The water maze device is circular in shape (1.8 m diameter) and tasked with locating a hidden platform (13 cm diameter) that was submerged 1 cm below the water’s surface. The pool was divided into four equal quadrants, with a hidden escape platform located in the center of one of the quadrants. Rats were given a navigation test for four consecutive days and placed at four different starting points during each trial [64]. Each trial was initiated by placing the rats in the water facing the wall to avoid observation before the trial, and the escape time was recorded. After rats navigated the hidden platform for escape, they were placed back into a cage, and the water was swapped to avoid hypothermia. For each phase, consisting of four trials, 120 s was the maximum time to escape. After four days of trial tasks, their behavior was recorded for 120 s, except for the hidden platform during the probe trial. Throughout this probe trial, the time spent and the number of crossings over the platform in the target quadrant were analyzed as an additional informative indicator of spatial learning and memory. A recording camera and animal movement tracking system followed their route and measured the individual swimming trajectories (Ethovision; Noldus Information Technology, Wageningen, The Netherlands) [65].

### 4.19. Statistical Analysis

All the data are described as mean ± standard error (SE). Repeated measure analysis of variance (ANOVA) was used to compare differences in mNSS scores and escape latency using SPSS software ver.21. Other comparisons between pairs of groups were performed with a two-tailed unpaired Student’s *t*-test. To compare the values in every four groups, raw data were analyzed by the Kruskal–Wallis test or one-way ANOVA with post hoc analysis using Bonferroni correction. *P*-values less than 0.05 (*p* < 0.05) indicated statistical significance.

## 5. Conclusions

Despite these limitations, the present study demonstrates BDNF-eMSCs’ efficacy in preventing neuronal death, neurological deficits, and cognitive impairment in a rodent model of TBI. Although our results suggest that BDNF-eMSCs may have significant clinical implications for treating TBI, additional preclinical studies are needed to develop effective therapeutic tools for humans.

## Figures and Tables

**Figure 1 pharmaceuticals-16-00436-f001:**
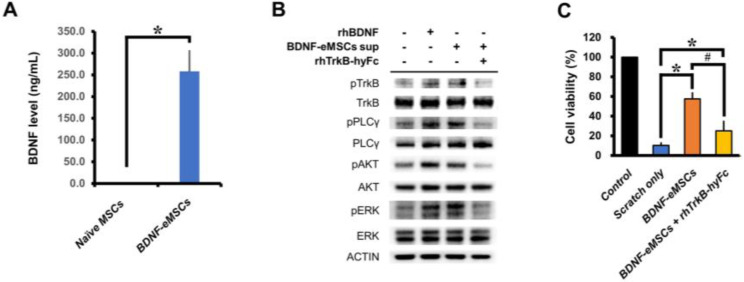
Characterization of BDNF-eMSCs. (**A**) BDNF secretion of naïve MSCs and BDNF-eMSCs measured by ELISA. Data are mean ± SEM; *n* = 3 per group. * *p* < 0.05 vs. naïve MSC group. (**B**) Representative immunoblot image showing activation of the BDNF–TrkB signaling pathway by treatment of supernatant of BDNF-eMSCs. (**C**) Analysis of cell viability by in vitro scratch assay after co-culture with BDNF-eMSC. Data are mean ± SEM; *n* = 4 per group. * *p* < 0.05 vs. scratch-only group; # *p* < 0.05 vs. BDNF-eMSC-treated group (One-way analysis of variance [ANOVA] followed by Bonferroni post hoc test: F = 39.54, *p* < 0.0001).

**Figure 2 pharmaceuticals-16-00436-f002:**
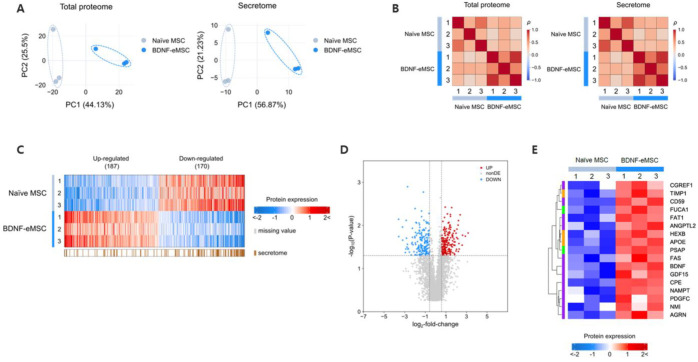
Proteomic analysis of BDNF-eMSC supernatant. (**A**) The principal component analysis (PCA) shows the first two principal components of data regarding their relationship in total proteome or secretome. (**B**) Correlation analysis shows replicates of the same pipeline cluster at both total proteome and secretome levels. (**C**) Heatmap of differentially expressed proteins (DEPs) in the total proteome (red and blue) and secretome (brown). (**D**) Volcano plot analysis shows DEPs of the total proteome. Red dots indicate significantly up-regulated genes, and blue dots indicate down-regulated genes. (**E**) Heatmap of significantly up-regulated proteins in secretome of BDNF-eMSCs compared to naïve MSCs. Analyzed DEPs are selected by BDNF-eMSCs enriched gene ontology biological process (GOBP) analysis: neurogenesis (purple), metabolism (green), or both (yellow).

**Figure 3 pharmaceuticals-16-00436-f003:**
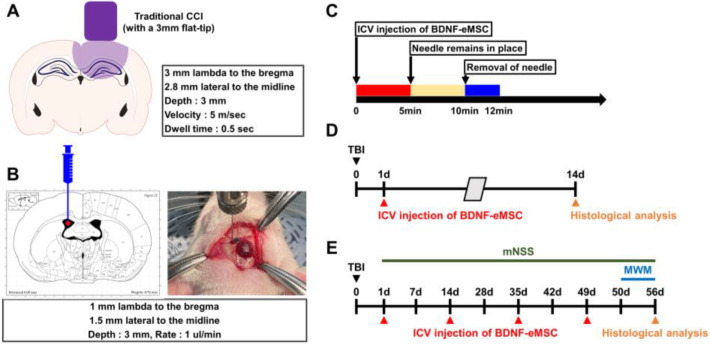
Schematic information of in vivo TBI experiments. (**A**) Schematic diagram of experimental condition for TBI using controlled cortical impact (CCI) device. (**B**,**C**) Method of intracerebroventricular injection of BDNF-eMSCs. BDNF-eMSCs were injected into the lateral ventricle region of the rat brain 24 h after TBI. (**D**,**E**) Timeline showing the experimental design of the sub-acute phase for 14 days (**D**) and the chronic phase for 56 days (**E**) post-TBI.

**Figure 4 pharmaceuticals-16-00436-f004:**
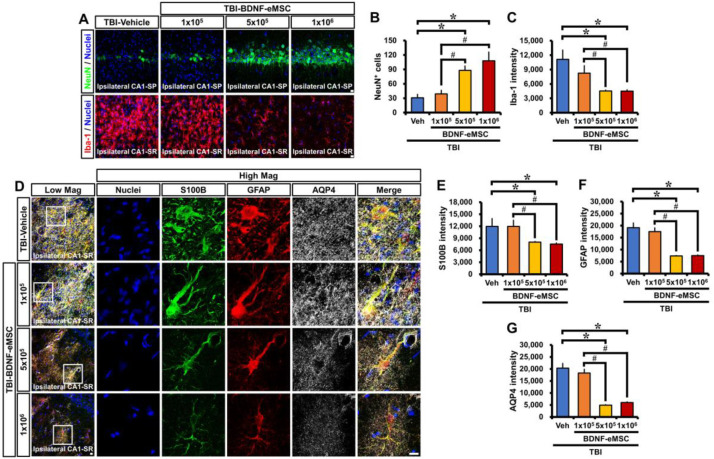
A single administration of BDNF-eMSCs reduces neuronal death and glial activation after TBI. (**A**) Representative images showing the expression of NeuN (green) or Iba-1 (red) in the CA1 stratum pyramidal (SP) or stratum radiatum (SR) of the ipsilateral hippocampus 14 days after TBI, respectively. Scale bar, 10 μm. (**B**,**C**) Quantification of the number of NeuN^+^ neurons (**B**) or immunofluorescence intensity of Iba-1 (**C**), as determined in the CA1 SP or SR of the ipsilateral hippocampus after TBI, respectively. Data are mean ± SEM; *n* = 4 per group. * *p* < 0.05 vs. vehicle-treated TBI group; # *p* < 0.05 vs. BDNF-eMSCs (1 × 10^5^)-treated TBI group (Kruskal–Wallis test followed by Bonferroni post hoc test: (**B**): Chi-square = 11.691, df = 3, *p* = 0.009; (**C**): Chi-square = 8.846, df = 3, *p* = 0.031). (**D**) Triple-label confocal micrographs of S100 calcium-binding protein B (S100B, green), glial fibrillary acidic protein (GFAP, red), and aquaporin-4 (AQP4, gray) in the CA1 SR of the ipsilateral hippocampus at 14 days after TBI. Scale bar, 10 μm. (**E**–**G**) Quantification of the immunofluorescence intensity of S100B (**E**), GFAP (**F**), or aquaporin-4 (**G**)**,** as determined in the CA1 SR of the ipsilateral hippocampus after TBI. Data are mean ± SEM; *n* = 4 per group. * *p* < 0.05 vs. vehicle-treated TBI group; # *p* < 0.05 vs. BDNF-eMSCs (1 × 10^5^)-treated TBI group (Kruskal–Wallis test followed by Bonferroni post hoc test: (**E**): Chi-square = 11.355, df = 3, *p* = 0.01; (**F**): Chi-square = 11.51, df = 3, *p* = 0.009; (**G**): Chi-square = 11.664, df = 3, *p* = 0.009).

**Figure 5 pharmaceuticals-16-00436-f005:**
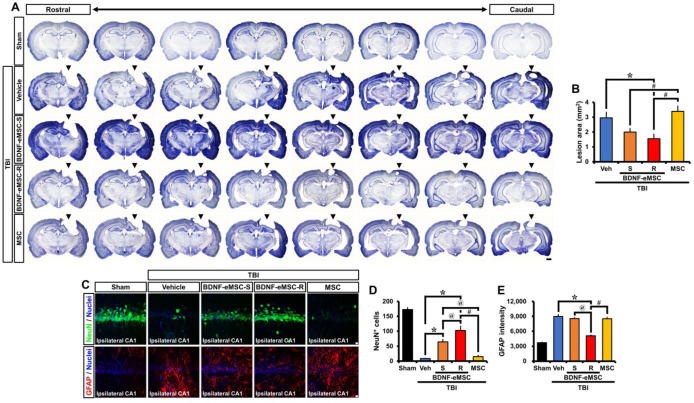
Repeated administration of BDNF-eMSCs not only reduces lesion area but also exerts neuroprotective effects after TBI. (**A**) Representative images of cresyl violet staining depicting coronal brain sections at rostral-caudal levels from −2.64 to −5.40 from the bregma. Scale bar, 1 mm. The black arrows represent the common damage site of TBI. (**B**) Graph showing the lesion area of the vehicle-, single BDNF-eMSC (BDNF-eMSC-S)-, repeated BDNF-eMSC (BDNF-eMSC-R)-, and MSC-treated groups at 8 weeks post-TBI. The lesion area is expressed in mm^2^. Data are mean ± SEM; *n* = 10 per group. * *p* < 0.05 vs. vehicle-treated TBI mice; # *p* < 0.05 vs. BDNF-eMSC-treated TBI mice (Kruskal–Wallis test followed by Bonferroni post hoc test: Chi-square = 13.225, df = 3, *p* = 0.004). (**C**) Representative images showing the expression of NeuN (green) or GFAP (red) in the CA1 from the ipsilateral hippocampus at 56 days post-TBI. Scale bar, 10 μm. (**D**,**E**) Quantification of the number of NeuN^+^ neurons (**D**) or immunofluorescence intensity of GFAP (**E**) as determined in the CA1 from the ipsilateral hippocampus after TBI. Data are mean ± SEM; *n* = 7–10 per group. * *p* < 0.05 vs. vehicle-treated group, # *p* < 0.05 vs. MSC-treated group, @ *p* < 0.05 vs. single BDNF-eMSC (BDNF-eMSC-S)-treated group (Kruskal–Wallis test followed by Bonferroni post hoc test: (**D**): Chi-square = 36.194, df = 4, *p* < 0.001; (**E**): Chi-square = 33.599, df = 4, *p* < 0.001).

**Figure 6 pharmaceuticals-16-00436-f006:**
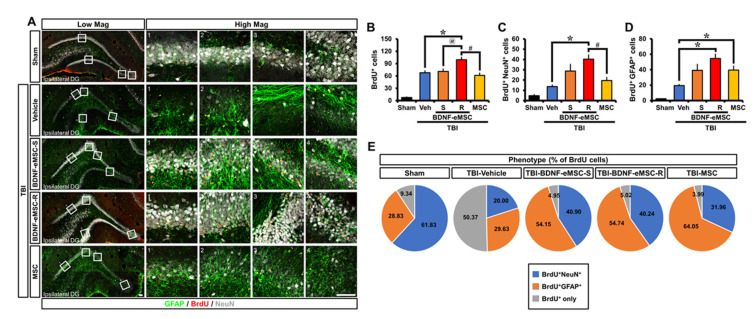
Repeated administration of BDNF-eMSCs increases adult hippocampal neurogenesis and gliogenesis after TBI. (**A**) Representative immunofluorescence images showing the phenotype of cells surviving in the sub-granular zone/granular cell layer (SGZ/GCL). Scale bar, 10 μm. (**B**–**D**) Quantification of the number of BrdU^+^ (**B**), BrdU^+^NeuN^+^ (**C**), or BrdU^+^GFAP^+^ cells (**D**), as determined in the SGZ/GCL from the ipsilateral dentate gyrus (DG) 56 days post-TBI. Data are mean ± SEM; *n* = 7–10 per group. * *p* < 0.05 vs. vehicle-treated group, # *p* < 0.05 vs. MSC-treated group, @ *p* < 0.05 vs. single BDNF-eMSC-treated group (Kruskal–Wallis test followed by Bonferroni post hoc test: (**B**): Chi-square = 24.64, df = 4, *p* < 0.001; (**C**): Chi-square = 25.278, df = 4, *p* < 0.001; (**D**): Chi-square = 26.462, df = 4, *p* < 0.001). (**E**) Phenotype distribution of newborn cells in the ipsilateral DG 8 weeks post-TBI. Data are mean ± SEM; *n* = 7–10 per group.

**Figure 7 pharmaceuticals-16-00436-f007:**
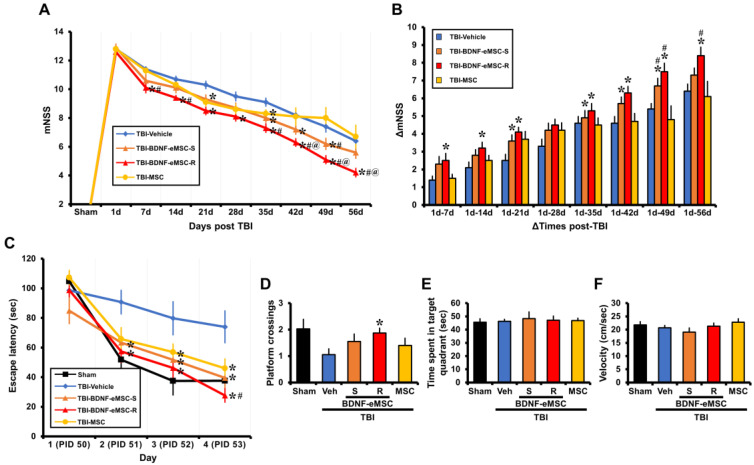
Behavioral effects of repeated administration of BDNF-eMSCs after TBI. (**A**) The mNSS was determined in rats for the entire experiment period after TBI. A score of 18 means that all tasks failed; a score of 0 indicates that all tasks were successfully completed. Data are mean ± SEM; *n* = 10 from each group, * *p* < 0.05 vs. vehicle-treated group, # *p* < 0.05 vs. MSC-treated group, @ *p* < 0.05 vs. single BDNF-eMSCs (BDNF-eMSC-S)-treated group (Repeated measure ANOVA; Day: F = 238.907, *p* < 0.001; Group: F = 8.806, *p* < 0.001; Day × Group interaction: F = 2.405, *p* < 0.001). (**B**) Changes in mNSS (ΔmNSS) were evaluated at various time intervals between day 1 and multiple pre-determined time points thereafter. Data are mean ± SEM; *n* = 10 from each group, * *p* < 0.05 vs. vehicle-treated group, # *p* < 0.05 vs. MSC-treated group (Repeated measures ANOVA; Time: F = 74.780, *p* < 0.001; Group: F = 12.041, *p* = 0.008; Day × Group interaction: F = 14.363, *p* < 0.001). (**C**) MWM performance. Escape latency of acquisition trial for 4 consecutive days starting post-injury day (PID) 50. Data are mean ± SEM; *n* = 10 from each group, * *p* < 0.05 vs. vehicle-treated group, # *p* < 0.05 vs. MSC-treated group (Repeated measure ANOVA; Day: F = 50.079, *p* < 0.001; Group: F = 12.455, *p* = 0.001; Day × group interaction: F = 2.752, *p* = 0.015). (**D**) MWM platform crossing. (**E**) Time spent in the target quadrant. (**F**) Velocity. Data are mean ± SEM; *n* = 10 from each group, * *p* < 0.05 vs. vehicle-treated group.

## Data Availability

Data is contained within the article and Appendix A.

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
