# Peer review of "Engineered Mesenchymal Stem Cells Over-Expressing BDNF Protect the Brain from Traumatic Brain Injury-Induced Neuronal Death, Neurological Deficits, and Cognitive Impairments"

_pharmaceuticals, 2023, doi:10.3390/ph16030436_

Round 1
Reviewer 1 Report
The present study evaluated the therapeutic potential of irradiated engineered human mesenchymal stem cells over-expressing brain-derived neurotrophic factor (BDNF) (BDNF-eMSCs) in protecting the brain against neuronal death, neurological deficits, and cognitive impairment in traumatic brain injury (TBI) rats. The study concluded that BDNF-eMSCs attenuate TBI-induced brain damage. This is a well-conducted study. The experimental design is presented in the text and Figure 3. Both histopathological and functional outcomes were measured up to 56 days post-TBI. Evaluating long-term outcomes is the main strength of the study. Minor comments below would help further strengthen the manuscript. Please provide details if the animals received any analgesics post-TBI surgery. Was the presence of BDNF-eMSCs during histopathological analysis? Please comment. The supplementary file has the label “Supplementary information for Antioxidants”. Since this manuscript is submitted to the journal Pharmaceuticals, please correct the name of the journal. Please provide a volcano plot for up and down-regulated proteins.
Author Response
Please provide details if the animals received any analgesics post-TBI surgery.
<Response: We appreciate this reviewer’s comment. In the present study, the animals did not receive any analgesics post-TBI surgery.>
Was the presence of BDNF-eMSCs during histopathological analysis? Please comment.
<Response: We appreciate this reviewer’s comment. We did not verify the presence of BDNF-eMSCs by histological analysis. After irradiation, we administered BDNF-eMSCs to the animals and expected them to present in one or two weeks. After BDNF-eMSCs were administered to normal mice, the residual amount was measured by qPCR. We found that BDNF-eMSCs disappeared from the brain within a week. The Discussion section of the revised manuscript includes limitations of the present study and future directions.>
The supplementary file has the label “Supplementary information for Antioxidants”. Since this manuscript is submitted to the journal Pharmaceuticals, please correct the name of the journal.
<Response: We corrected it.>
Please provide a volcano plot for up and down-regulated proteins.
<Response: We appreciate this reviewer’s comment. We added a “volcano plot” for up- and down-regulated proteins to Figure 2D.>

Reviewer 2 Report
The authors analyzed the therapeutical potential of BDGF expressing engineered-human mesenchymal stem cells on the recovery of rats after TBI (CCI). After single or multiple ICV injections of BDNF-eMSCs there is improved recovery from CCI, evident by reduced neuronal death, reduced glial activation and improved neurological and cognitive functions.
Overall, the presented data support the author’s conclusions, but several experimental details should be clarified before publication.
- 2.1. Cell preparation: Please clarify that the eMSCs used are of human origin. Also clarify how the irradiated BDNF-eMSCs were kept before ICV injection (were they irradiated and frozen until injection or were freshly maintained cultures irradiated the day of the injection). The levels of radiation are high, so I wonder about the survival of these cells after radiation. Experimental details or references about this should be included in the Methods.
- It is unclear in the in vitro experiments (Figure 1 and 2) if cells were irradiated or not.
- There is a contradiction in how the Figure 1C experiment is explained in the figure legend and in the text. The figure legend states that analysis of viability by in vitro scratch assay was performed after treatment with BDNF-eMSCs supernatant but the text reads, “In the group of primary neurons co-cultured with BDNF-eMSCs (1:1 ratio to neuron) cell viability was increased”. If the latter is true, how do you discard that the increase viability is due to the MSCs presence rather than the neuronal survival?
- For the proteomic analysis experiments, is important to outline the details of how the BDNF-eMSCs and naïve supernatants were prepared so the data can be compared between laboratories. What is the formulation of the conditional media used? Were irradiated cells use for the experiment? For how many days were the cells maintained in culture? Which passage was used?
- The title of Figure 2 is confusing. “Proteomic analysis of BDNF-eMSC supernatant” appears to indicate that only the secretome was analyzed but you are showing total proteome data as well, what leads me to believe that cells were also analyzed. Is this correct? Please clarify in materials and methods as well.
- In Figure 4A, pics for NeuN and Iba1 seem to be in different areas based on the nuclei staining. Please clarify on the figure legend or show a low magnification figure like the ones in Fig6A to orient the viewer.
- Also, in Figure 4D pics don’t show the CA1 nuclei layer, so where were these pictures taken? How was the signal intensity analyzed? By equivalent area, as a percentage of the area or per positive cell?
- In Figure 6B is unclear if the positive cells are the sum of the whole gyrus or if they represent the number of cells in a fixed area.
- Figure 7 showed improvement on mNSS even in TBI-MSC group at 35 days and the scape latency of the MWM at all ages in TBI-MSC. I believe this is an important point to discuss since it appears to indicate that just naïve cells have limited therapeutic potential and that BDNF is only partially responsible for the animals’ cognitive recovery. Please include a comment in the results or discussion section.
Minor corrections
Abstract 4th sentence: “ BDNF-mess” should read “BDNF-eMSCs”
Define the followed abbreviations in the text: mNSS, MWM, CCI
Author Response
2.1. Cell preparation: Please clarify that the eMSCs used are of human origin. Also clarify how the irradiated BDNF-eMSCs were kept before ICV injection (were they irradiated and frozen until injection or were freshly maintained cultures irradiated the day of the injection). The levels of radiation are high, so I wonder about the survival of these cells after radiation. Experimental details or references about this should be included in the Methods.
<Response: We appreciate this reviewer’s comments. In the present study, we used engineered human mesenchymal stem cells over-expressing BDNF. BDNF-eMSCs were irradiated while frozen and stored in the LN2 vapor phase under freezing conditions until used for in vitro and in vivo experiments. BDNF-eMSCs were thawed, prepared through a concentration procedure, and kept at 2 ~ 8℃ until used. As the reviewer suggested, we added these details to the Materials and Methods section of the revised manuscript.>
It is unclear in the in vitro experiments (Figure 1 and 2) if cells were irradiated or not.
<Response: We appreciate this reviewer’s comment. Irradiated cells are denoted as BDNF-eMSCs in Materials and Methods Section 2.1. Cell preparation. We used irradiated BDNF-eMSCs in our in vitro and in vivo experiments except for long-term culture experiments in supplementary data.>
There is a contradiction in how the Figure 1C experiment is explained in the figure legend and in the text. The figure legend states that analysis of viability by in vitro scratch assay was performed after treatment with BDNF-eMSCs supernatant but the text reads, “In the group of primary neurons co-cultured with BDNF-eMSCs (1:1 ratio to neuron) cell viability was increased”. If the latter is true, how do you discard that the increase viability is due to the MSCs presence rather than the neuronal survival?
<Response: The in vitro experiment shown in Figure 1C was conducted using a trans-well with a pore size that allows proteins to pass through but not cells. Therefore, increased cell viability by proteins, including secreted BDNF, was confirmed. We revised our paper accordingly and felt that this reviewer’s comments helped clarify and improve our work.>
For the proteomic analysis experiments, is important to outline the details of how the BDNF-eMSCs and naïve supernatants were prepared so the data can be compared between laboratories. What is the formulation of the conditional media used? Were irradiated cells use for the experiment? For how many days were the cells maintained in culture? Which passage was used?
<Response: We appreciate this reviewer’s comments. To obtain the cell supernatant for proteomic analysis, naïve MSCs and BDNF-eMSCs were seeded with 1 × 106 cells into T175 flasks. After 3 days, the complete media was changed to phenol-free DMEM (WELGENE, Gyeongsan, South Korea) and cultured for 24 hours without fetal bovine serum. BDNF-eMSCs were used after irradiation under the same conditions as other in vitro and in vivo experiments. However, naïve MSCs were not irradiated for secretome analysis. The passage of naïve MSCs was P5 and BDNF-eMSCs was P45.>
The title of Figure 2 is confusing. “Proteomic analysis of BDNF-eMSC supernatant” appears to indicate that only the secretome was analyzed but you are showing total proteome data as well, what leads me to believe that cells were also analyzed. Is this correct? Please clarify in materials and methods as well.
<Response: We appreciate this reviewer’s comments. Proteomic analysis was performed using the supernatant of naive MSCs and BDNF-eMSCs. During the preparation of supernatant samples, non-secreted proteins may have been included. Therefore, all the identified proteins form the total proteome. Only the proteins reported as secreted proteins in the reference database were included in the secretome. The Materials and Methods section of the revised manuscript was modified to reflect this reviewer’s comments.>
In Figure 4A, pics for NeuN and Iba1 seem to be in different areas based on the nuclei staining. Please clarify on the figure legend or show a low magnification figure like the ones in Fig6A to orient the viewer.
<Response: We appreciate this reviewer’s comment. Figure 4A shows images of NeuN or Iba-1 in the stratum pyramidale (SP) or stratum radiatum (SR) of ipsilateral CA1, respectively. We corrected the Figure 4 legend of the revised manuscript accordingly.>
Also, in Figure 4D pics don’t show the CA1 nuclei layer, so where were these pictures taken? How was the signal intensity analyzed? By equivalent area, as a percentage of the area or per positive cell?
<Response: We appreciate this reviewer’s comments. Figure 4D shows images of the ipsilateral CA1’s stratum radiatum (SR). Immunofluorescence signals were detected using a Zeiss LSM 710 confocal microscope (Carl Zeiss, Oberkochen, Germany) in sequential scanning mode for DAPI and Alexa 488, 594, and 647. Image stacks (1024 × 1024 pixels) from consecutive slices of 0.5–0.8 μm in thickness were obtained by averaging fifteen scans per slice, which were processed using ZEN 2 blue edition (Carl Zeiss). The mean intensity for Iba-1, S100B, GFAP, or AQP4 was quantified by the equivalent area using ZEN 2 blue edition (Carl Zeiss). We corrected these details in the revised manuscript.>
In Figure 6B is unclear if the positive cells are the sum of the whole gyrus or if they represent the number of cells in a fixed area.
<Response: Figure 6B-D shows five coronal sections (330 μm intervals) collected from each animal, ranging from 2.92 to 4.56 mm of bregma. A blinded observer analyzed fluorescence images to count the number of BrdU+, BrdU+ GFAP+, or BrdU+ NeuN+ cells in the fixed area of hippocampal DG from the ipsilateral hemisphere. Regions of interest were drawn manually over the granular cell layer (GCL) and SGZ along the superior and inferior blades of the DG. the number of immunopositive cells in each section was counted for each animal, and the result is expressed as the mean number of immunopositive cells per mm2 ± SEM. The revised manuscript mentions the above methods.>
Figure 7 showed improvement on mNSS even in TBI-MSC group at 35 days and the escape latency of the MWM at all ages in TBI-MSC. I believe this is an important point to discuss since it appears to indicate that just naïve cells have limited therapeutic potential and that BDNF is only partially responsible for the animals’ cognitive recovery. Please include a comment in the results or discussion section.
<Response: We added this suggested description to the Results section of the revised manuscript.>
Minor corrections
Abstract 4th sentence: “ BDNF-mess” should read “BDNF-eMSCs”
<Response: We corrected this in the revised manuscript.>
Define the followed abbreviations in the text: mNSS, MWM, CCI
<Response: We corrected this in the revised manuscript.>

Reviewer 3 Report
In this manuscript, the authors evaluate the therapeutic potential of irradiated engineered human mesenchymal stem cells overexpressing brain-derived neurotrophic factor. Overall, this manuscript is very well presented with sound study design as demonstrated in the methodology. Apart from English editing by a professional to better deliver the information. This manuscript is suitable for publication. However, i encourage the authors to thoroughly discuss their results in comparison to available literature and to clearly describe their conclusion with reference to limitations and future recommendations.
Author Response
In this manuscript, the authors evaluate the therapeutic potential of irradiated engineered human mesenchymal stem cells overexpressing brain-derived neurotrophic factor. Overall, this manuscript is very well presented with sound study design as demonstrated in the methodology. Apart from English editing by a professional to better deliver the information. This manuscript is suitable for publication. However, I encourage the authors to thoroughly discuss their results in comparison to available literature and to clearly describe their conclusion with reference to limitations and future recommendations.
<Response: We appreciate this reviewer’s comments and agree with this reviewer’s point. Thus, we added the limitations of the present study and future directions to the Discussion and Conclusion with references. “This study has several potential limitations that must be addressed in future studies. (1) In the present study, we did not verify the presence of BDNF-eMSCs by histological analysis. After irradiation, BDNF-eMSCs were administered to animals, and we expected them to present within one or two weeks. After BDNF-eMSCs were administered to normal mice, the residual amount was measured by qPCR. We found that BDNF-eMSCs disappeared from the brain within a week. It would be interesting to explore the presence of BDNF-eMSCs after a longer period after administration. (2) Exploring the mechanism of BDNF-eMSCs after TBI may be a future direction for human translation. Despite these limitations, the present study demonstrates BDNF-eMSCs’ efficacy in preventing neuronal death, neurological deficits, and cognitive impairment in a rodent model of TBI. Although our study results suggest that BDNF-eMSCs may have significant clinical implications for treating TBI, additional preclinical studies are needed to develop effective therapeutic tools for humans.>
